analytical chemistry

synchronous fluorimetry, bambuterol, montelukast, International Council of Harmonization guidelines

**Author for correspondence:**
Samah Abo El Abass
e-mail: dr_samah157@yahoo.com

This article has been edited by the Royal Society of Chemistry, including the commissioning, peer review process and editorial aspects up to the point of acceptance.

# Quick simultaneous analysis of bambuterol and montelukast based on synchronous spectrofluorimetric technique

Rania El Gamal[1,2], Samah Abo El Abass[1] and Heba M. Elmansi[1]

[1]Department of Pharmaceutical Analytical Chemistry, Faculty of Pharmacy, Mansoura University, Mansoura 35516, Egypt
[2]Pharmaceutical Chemistry Department, College of Pharmacy, Prince Sattam Bin-Abdul Aziz University, PO Box 173, Al-Kharj 11942, Kingdom of Saudi Arabia

REG, 0000-0002-0036-1637; SAEA, 0000-0002-9073-9903; HME, 0000-0002-3953-7169

Sensitive, simple and green analytical methodology for simultaneous estimation of bambuterol and montelukast as a combined medication based on their native fluorescence character was developed. The method relies on synchronous spectrofluorimetry to solve the problem of the overlapping emission spectra of the studied drugs. Using second derivative synchronous spectra enabled the simultaneous quantitation of bambuterol and montelukast without interference. The peak amplitudes of the aqueous solutions at $\Delta\lambda = 20$ nm were estimated at 284 and 304 nm for bambuterol and at 374 and 384 nm for montelukast. A linear relationship was achieved over the concentration range of 0.2–1.00 µg ml$^{-1}$ for bambuterol and 0.4–2.00 µg ml$^{-1}$ for montelukast. All factors and parameters were carefully studied to obtain the highest sensitivity and good precision of the proposed method. Additionally, the validation criteria were assessed in accordance with International Council of Harmonization (ICH) guidelines. The method was used for the estimation of both drugs in their raw materials, synthetic mixtures as well their combined tablets with good agreement between its results and those from the comparison method.

## 1. Introduction

Bambuterol hydrochloride (BMB) (figure 1) is chemically (RS)-5-(2-tert-Butylamino-1-hydroxyethyl)-m-phenylene bis(dimethyl-

**Figure 1.** Structural formula of studied drugs. (a) Bambuterol hydrochloride (BMB) and (b) montelukast sodium (MTK).

carbamate) hydrochloride. It is a long acting bronchodilator used for airways obstruction as in case of asthma due to its direct-acting sympathomimetic and beta$_2$ agonist action [1]. BMB is an official drug in British Pharmacopoeia [2] and has been determined by several methods, such as spectrophotometry [3–6], spectrofluorimetry [7–9], electrochemical methods [10,11] and HPLC [12–15].

Montelukast sodium (MTK) (figure 1), is selective leukotriene receptor antagonist named as: sodium1-[({(R)-m-[(E)-2-(7-chloro-2-quinolyl)-vinyl]-α-[o-(1-hydroxy-1 methylethyl)phenethyl]benzyl}thio) methyl] cyclopropane acetate (figure 1). It is used in the maintenance treatment of asthma [1]. Different analytical methods for determination of MTK were developed, including spectrophometric methods [16–21], fluorimetry [22], TLC [23,24], HPLC [25–31], capillary electrophoresis [32] and voltammetry [33].

BMB and MTK are present in combined tablet formulations with a pharmaceutical ratio of 1 : 1 to treat chronic asthma, bronchospasm and reversible airway obstruction. The literature includes different methods for simultaneous analysis of BMB and MTK, such as HPLC [30,34] and spectrophotometry [35,36]. The reported methods suffered from using hazardous toxic solvents such as acetonitrile and methanol [34–36] in addition to long runs up to 20 min [30].

Fluorescence measurements are typically based on the presence of aromatic molecules. An important feature of the fluorimetric technique is high sensitivity detection [37]. This technique has several advantages besides its simplicity, including low cost, availability, short analysis time in addition to the absence of sample pretreatment or multi-step analysis. Fluorescence spectroscopy has wide applications in the field of pharmaceutical analysis. However, multi-component analysis is considerably problematic because of overlapping spectra, so that resolving mixtures of drugs is unsatisfactory. This problem of reduced selectivity could be overcome by synchronous fluorimetry [38]. Furthermore, combining synchronous scanning spectrofluorimetry with derivative techniques provides higher recovery percentages at lower concentration levels, higher selectivity and overcomes baseline variation problems [38].

Accordingly, a simple new method was investigated in order to analyse both BMB and MTK simultaneously without interference from each other relying on derivative synchronous spectrofluorimetry. Water was used as a diluting solvent which gives advantages of the proposed method of being green and cost effective.

# 2. Experimental set-up

## 2.1. Instruments

— A FP-8200 JASCO Spectrofluorophotometer (Japan) with a Xe arc lamp was used. Synchronous fluorescence spectra were recorded by scanning both monochromators at $\Delta\lambda = 20$ nm and a scan rate of 600 nm min$^{-1}$ using 10 nm slit width. Fluorescence data manager software was used to obtain the derivative spectra. JASCO Spectra Manager$^{TM}$ CFR, for FP-800 series, Copyright 2011, JASCO, Tokyo, Japan.

— Shimadzu LC-20AD Prominence liquid chromatograph equipped with an SIL-20 AD auto sampler and a SPD-20A UV detector was used to apply the comparison method.

— Seven2Go S2. -Basic; pH mV$^{-1}$ portable meter, Mettler Toledo (USA), was used for pH measurements.

— Ultrasonic bath, model Branson 2800.

— A&D GR300 analytical balance.

**Table 1.** Analytical performance data for the determination of BMB and MTK by the proposed method.

| parameter | BMB | | MTK | |
|---|---|---|---|---|
| | at 284 nm | at 304 nm | at 374 nm | at 384 nm |
| linearity range ($\mu g\ ml^{-1}$) | 0.2–1.0 | 0.2–1.0 | 0.4–2.0 | 0.4–2.0 |
| intercept (a) | −0.244 | 0.062 | −0.013 | 0.074 |
| slope (b) | 4.103 | 1.563 | 0.219 | 0.241 |
| correlation coefficient (r) | 1.0000 | 0.9999 | 0.9998 | 0.9999 |
| s.d. of residuals ($S_{y/x}$) | 0.012 | 0.008 | 0.003 | 0.003 |
| s.d. of intercept ($S_a$) | 0.012 | 0.008 | 0.004 | 0.003 |
| s.d. of slope ($S_b$) | 0.019 | 0.012 | 0.003 | 0.002 |
| s.d. | 0.45 | 0.86 | 1.24 | 1.03 |
| % RSD[a] | 0.450 | 0.864 | 1.235 | 1.032 |
| % error[b] | 0.201 | 0.386 | 0.553 | 0.462 |
| LOD ($\mu g\ ml^{-1}$)[c] | 0.010 | 0.017 | 0.052 | 0.040 |
| LOQ ($\mu g\ ml^{-1}$)[d] | 0.030 | 0.050 | 0.158 | 0.122 |

[a]Percentage relative standard deviation.
[b]Percentage relative error.
[c]Limit of detection.
[d]Limit of quantitation.

## 2.2. Materials and chemicals

The chemicals used throughout the study were of analytical reagent grade and the solvents were of HPLC grade.

— Bambuterol was kindly provided by AstraZeneca pharmaceutical, Egypt. Montelukast was kindly provided by Amoun Pharmaceutical Company S.A.E., Egypt.
— Montec® Plus tablets, product of SUN PHARMA (India), labelled to contain 10 mg of BMB and 10 mg of MTK per tablet, Batch No: EMT 1859.
— Methanol and acetonitrile products of Sigma-Aldrich, Steinheim, France.
— Glacial acetic acid product of BDH laboratory supplies, Poole, England.
— Ethanol absolute and acetone products of Scharlab, Spain.
— Sodium acetate trihydrate and sodium dodecyl sulphate were obtained from Lobachemie, Mumbai, India.
— Boric acid and sodium hydroxide were purchased from central drug house (CDH), New Delhi, India.
— Tween 80, methyl cellulose and borax were purchased from Assagaf pharma, Saudi Arabia.

## 2.3. Standard solutions, buffers and surfactants

BMB and MTK stock solutions were prepared in methanol in the concentration of $100\ \mu g\ ml^{-1}$. MTK stock solution was protected from light by wrapping in aluminium foil.

Acetate buffer ($0.2\ mol\ l^{-1}$, pH 3.6–5.6) was prepared by mixing appropriate volumes of 0.2 M acetic acid and 0.2 M sodium hydroxide.

Borate buffer ($0.2\ mol\ l^{-1}$, pH 6.5–9.0) solutions were freshly prepared by mixing appropriate volumes of 0.2 M boric acid and 0.2 M borax.

Sodium dodecyl sulfate (SDS), methylcellulose and Tween 80 were prepared as 0.1% w/v aqueous solutions.

## 2.4. General procedures

### 2.4.1. Construction of calibration curves

Aliquots of each of BMB and MTK standard solutions within the studied concentration ranges (table 1) were transferred into a set of 10 ml volumetric flasks, completed with distilled water and mixed. A blank

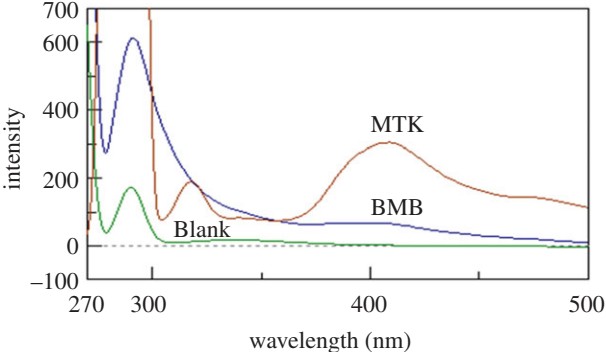

**Figure 2.** Overlapping emission spectra of bambuterol and montelukast after excitation at 263 and 286 nm, respectively.

experiment was carried out concurrently. The synchronous fluorescence spectra were recorded at $\Delta\lambda = 20$ nm and second derivative synchronous fluorescence spectra were then derived. To obtain smooth spectra and to reach the highest sensitivity, 25 points were used for deriving the derivative synchronous fluorescence spectra. The peak amplitudes were measured at 284 and 304 nm for BMB and at 374 and 384 nm for MTK. A plot of the peak amplitudes against drug concentration in µg ml$^{-1}$ was constructed to obtain the calibration graphs and the corresponding regression equations were derived.

### 2.4.2. Analysis of synthetic mixtures of BMB and MTK

Aliquots of BMB and MKT standard solutions in the ratio of (1 : 1), as in their formulated tablets, were transferred into a 10 ml volumetric flask series, completed with distilled water and mixed well. The mixtures were analysed as described before to calculate the percentage recoveries by referring to the corresponding regression equation.

### 2.4.3. Analysis of combined tablet dosage forms

The contents of 10 powdered tablets were ground and mixed well. An amount equivalent to 10.0 mg of BMB and MTK was transferred into a 100 ml volumetric flask and about 40.0 ml of methanol were added. The solutions were sonicated for 30 min, completed to the mark with methanol and filtered through cellulose acetate syringe filter. Aliquots were then taken to be assayed by the general procedure. The contents of tablets were computed with the aid of the corresponding regression equations.

## 3. Results and discussion

Both BMB and MTK were reported to exhibit native fluorescence. BMB was determined by conventional spectrofluorometry in water at 263/298 nm [7]. Meanwhile, MTK was determined in methanol at 390 nm using 340 nm for excitation [22]. Scanning the fluorescence spectra of the studied drugs at their optimum excitation/emission revealed that their spectra suffered from significant overlapping that hinders their simultaneous determination (figure 2).

Synchronous fluorescence spectroscopy at a suitable $\Delta\lambda$ can increase selectivity in case of multi-component mixtures analysis [17]. Figures 3 and 4 show the synchronous spectra of different concentrations of BMB in the presence of MKT and different concentrations of MTK in the presence of BMB, respectively. There is still a degree of interference from MTK spectra in case of measuring BMB, but MTK could be measured in the presence of BMB. Accordingly, we tried the derivative synchronous technique and the spectra of BMB and MTK were resolved using the second derivative amplitudes. Figures 5 and 6 show that BMB can be determined at 284 and 304 nm with MTK present, and MTK could be determined at 374 and 384 nm in the presence of BMB.

### 3.1. Optimization of different parameters

#### 3.1.1. Effect of $\Delta\lambda$ value

The separation and resolution of fluorescence spectra could be greatly influenced by the choice of $\Delta\lambda$. A study for $\Delta\lambda$ from 20 to 160 nm was performed. It was found that $\Delta\lambda$ of 20 nm gave rise to acceptable sensitivity and lower spectral interference.

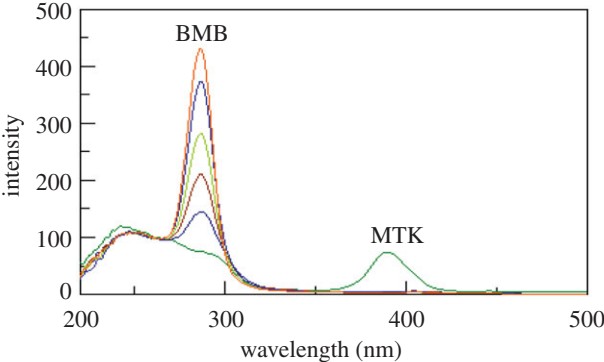

**Figure 3.** Synchronous fluorescence spectra of different concentrations of bambuterol (0.2, 0.4, 0.6, 0.8 and 1 µg ml$^{-1}$) in the presence of 1.0 µg ml$^{-1}$ montelukast.

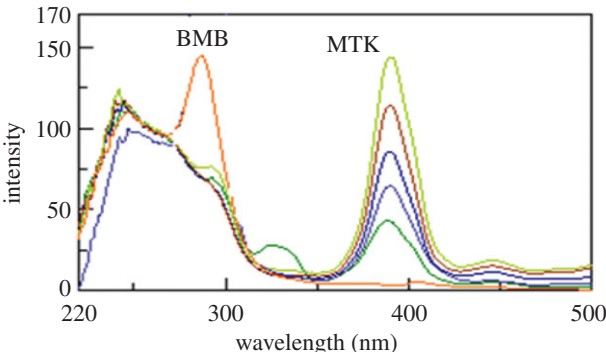

**Figure 4.** Synchronous fluorescence spectra of different concentrations of montelukast (0.4, 0.8, 1.2, 1.6 and 2.0 µg ml$^{-1}$) in the presence of 0.2 µg ml$^{-1}$ bambuterol.

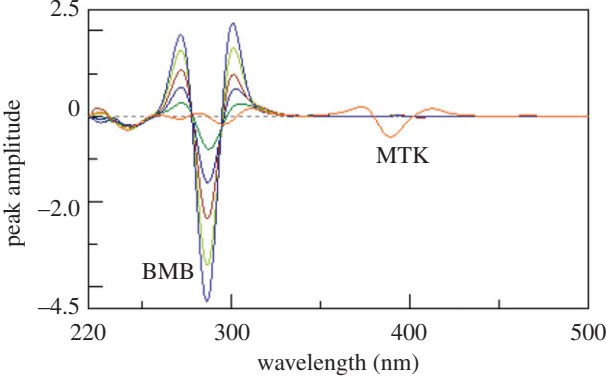

**Figure 5.** Second derivative synchronous fluorescence spectra of different concentrations of bambuterol (0.2, 0.4, 0.6, 0.8 and 1 µg ml$^{-1}$) in the presence of 1.0 µg ml$^{-1}$ montelukast.

### 3.1.2. Effect of pH

The effect of pH on the spectra of the BMB and MTK was studied. The pH range of 3.6–5.6 was investigated using 0.2 M acetate buffer and the pH range of 6.5–9.3 was examined using 0.2 M borate buffer (figure 7). It was noted that maximum intensity for BMB was obtained at pH 7.5, whereas for MTK, maximum synchronous fluorescence intensity (SFI) was at the pH range of 8–9; for both drugs buffer did not enhance the SFI if compared with not using buffer at all. Hence, no buffer was used in the analysis procedures.

### 3.1.3. Effect of different surfactants

Different surfactants in the concentration of 0.1% (w/v) were tried to evaluate its effect on the synchronous spectra shape or sensitivity. CMC, sodium dodecyl sulfate and Tween 80 were

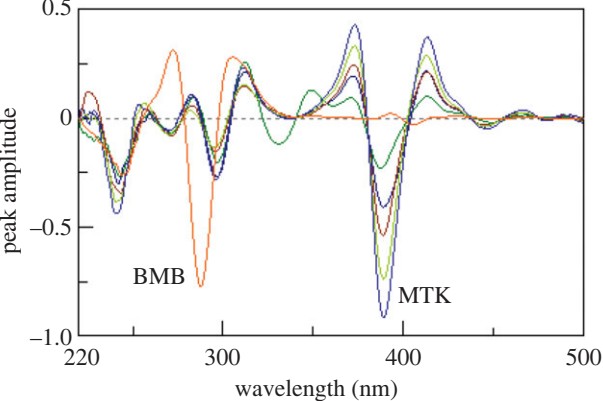

**Figure 6.** Second derivative synchronous fluorescence spectra of different concentrations of montelukast (0.4, 0.8, 1.2, 1.6 and 2.0 µg ml$^{-1}$) in the presence of 0.2 µg ml$^{-1}$ bambuterol.

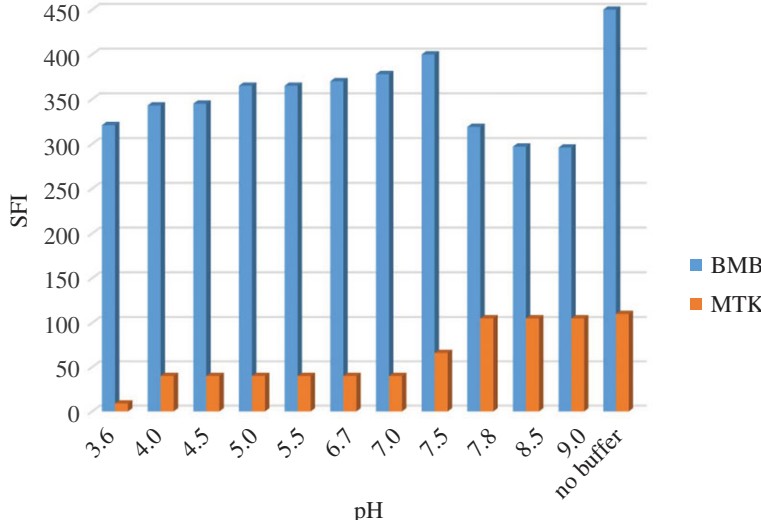

**Figure 7.** Effect of pH on the synchronous fluorescence intensity of montelukast and bambuterol (concentration of each 1.0 µg ml$^{-1}$).

investigated. It was found that CMC did not affect the SFI of either drug, whereas SDS decreased the SFI of BMB. Tween 80 resulted in a pronounced increase in synchronous fluorescence of both drugs at 388 nm; however, it potentially affected the resolution of peaks by derivative synchronous technique. Therefore, to achieve optimal resolution and to determine each of the studied drugs simultaneously, none of the studied surfactant was used.

### 3.1.4. Effect of different solvents

Diluting solvents were investigated to enhance the shape of spectra, intensity and resolution. Methanol, ethanol, acetonitrile, acetone and water were tried (figure 8). The fluorescence of BMB is highest in water, while ethanol was the best for MTK. However, it resulted in greater peak overlap and showed high blank reading at BMB fluorescence maximum. Therefore, for simplicity and convenience, water was the solvent of choice.

## 3.2. Validation parameters

International Council of Harmonization (ICH) guidelines [39] were followed to validate the proposed method as follows:

The measured second derivative synchronous amplitude was linear with the concentration over the range of 0.2–1.00 µg ml$^{-1}$ for BMB and 0.4–2.00 µg ml$^{-1}$ for MTK. The calculated parameters including correlation coefficient, slope and intercept are summarized in table 1.

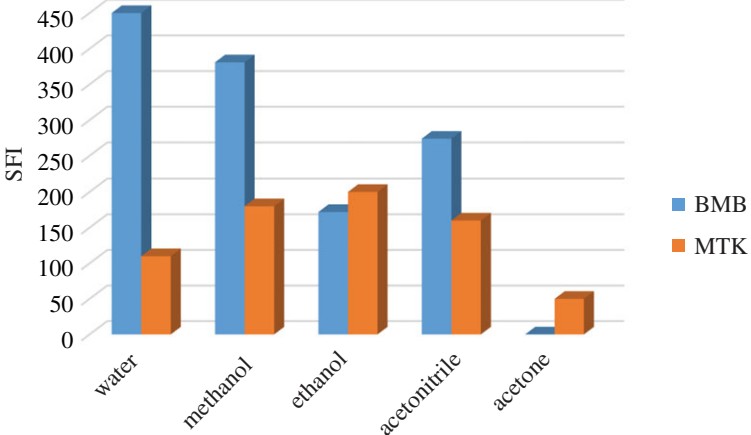

**Figure 8.** Effect of different diluting solvents on the synchronous fluorescence intensity of montelukast and bambuterol (concentration of each 1.0 µg ml$^{-1}$).

**Table 2.** Assay results for the determination of the BMB and MTK in pure form by the proposed and comparison method. The figures between parentheses are the tabulated *t*- and *F*-values at $p = 0.05$ [40].

| compound | proposed method | | | | | comparison method [29] |
|---|---|---|---|---|---|---|
| | | at 284 nm | | at 304 nm | | |
| | conc. taken (µg ml$^{-1}$) | conc. found (µg ml$^{-1}$) | % found | conc. found (µg ml$^{-1}$) | % found | % found |
| BMB | 0.2 | 0.201 | 100.45 | 0.197 | 98.55 | 100.42 |
| | 0.4 | 0.401 | 100.20 | 0.402 | 100.48 | 99.34 |
| | 0.6 | 0.596 | 99.30 | 0.600 | 100.05 | 100.32 |
| | 0.8 | 0.802 | 100.30 | 0.806 | 100.71 | |
| | 1.0 | 1.000 | 100.01 | 0.995 | 99.51 | |
| mean | | | 100.05 | | 99.86 | 100.03 |
| ± s.d. | | | 0.45 | | 0.86 | 0.60 |
| *t*-test | | | 0.068 (2.44)[a] | | 0.29 (2.44)[a] | |
| *F*-test | | | 1.76 (6.94)[a] | | 2.09 (19.25)[a] | |
| compound | proposed method | | | | | comparison method [29] |
| | | at 284 nm | | at 304 nm | | |
| | conc. taken (µg ml$^{-1}$) | conc. found (µg ml$^{-1}$) | % found | conc. found (µg ml$^{-1}$) | % found | % found |
| MTK | 0.4 | 0.400 | 100.00 | 0.398 | 99.50 | 99.21 |
| | 0.8 | 0.814 | 101.71 | 0.813 | 101.58 | 101.23 |
| | 1.2 | 1.179 | 98.25 | 1.186 | 98.81 | 99.41 |
| | 1.6 | 1.599 | 99.94 | 1.598 | 99.85 | |
| | 2.0 | 2.007 | 100.36 | 2.006 | 100.28 | |
| mean | | | 100.05 | | 100.00 | 99.95 |
| ± s.d. | | | 1.24 | | 1.03 | 1.11 |
| *t*-test | | | 0.12 (2.44)[a] | | 0.069 (2.44)[a] | |
| *F*-test | | | 1.23 (19.25)[a] | | 1.16 (6.94)[a] | |

[a]N.B. Each result is the average of three separate determinations.

**Table 3.** Accuracy and precision data for the determination of BMB and MTK by the proposed method.

| parameter | | at 284 nm | | | at 304 nm | | | at 374 nm | | | at 384 nm | | |
| --- | --- | --- | --- | --- | --- | --- | --- | --- | --- | --- | --- | --- | --- |
| (µg ml⁻¹) | | 0.4 | 0.8 | 1.0 | 0.4 | 0.8 | 1.0 | 0.4 | 0.8 | 1.2 | 0.4 | 0.8 | 1.2 |
| intra-day | mean | 99.27 | 100.11 | 99.73 | 99.58 | 98.91 | 100.32 | 99.11 | 99.04 | 98.57 | 97.95 | 100.49 | 101.02 |
| | s.d. | 1.14 | 1.77 | 0.61 | 0.66 | 0.84 | 0.79 | 0.91 | 1.50 | 0.79 | 0.3 | 1.14 | 0.74 |
| | % RSD | 1.15 | 1.77 | 0.61 | 0.66 | 0.84 | 0.78 | 0.91 | 1.51 | 0.80 | 0.3 | 1.14 | 0.73 |
| | % error | 0.66 | 1.02 | 0.35 | 0.38 | 0.49 | 0.45 | 0.53 | 0.87 | 0.46 | 0.17 | 0.66 | 0.42 |
| inter-day | mean | 99.02 | 100.15 | 99.01 | 100.07 | 99.59 | 99.82 | 98.84 | 98.83 | 99.68 | 99.53 | 99.10 | 99.33 |
| | s.d. | 0.90 | 1.74 | 1.05 | 0.55 | 0.81 | 1.02 | 1.08 | 0.87 | 0.62 | 0.68 | 0.64 | 0.68 |
| | % RSD | 0.91 | 1.73 | 1.06 | 0.55 | 0.81 | 1.02 | 1.09 | 0.88 | 0.62 | 0.68 | 0.65 | 0.68 |
| | % error | 0.53 | 1.00 | 0.61 | 0.32 | 0.47 | 0.59 | 0.63 | 0.51 | 0.36 | 0.39 | 0.37 | 0.40 |

[a]N.B. Each result is the average of three separate determinations.

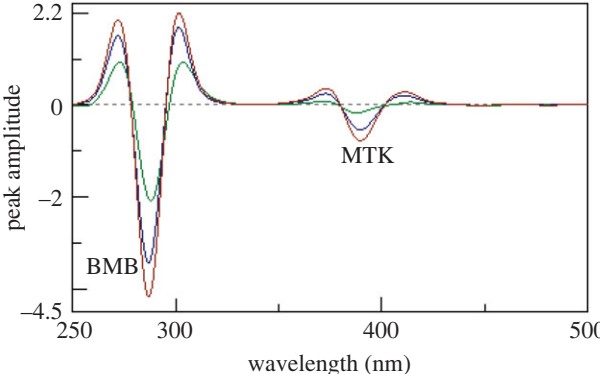

**Figure 9.** Second derivative synchronous fluorescence spectra of different concentrations of synthetic mixtures of montelukast and bambuterol (0.4, 0.8 and 1.0 µg ml$^{-1}$).

**Table 4.** Assay results for the determination of the studied drugs in their combined tablet by the proposed and comparison method. The figures between parentheses are the tabulated $t$- and $F$-values at $p = 0.05$ [40].

| compound | proposed method | | | | | comparison method [29] |
|---|---|---|---|---|---|---|
| | | at 284 nm | | at 304 nm | | |
| | conc. taken (µg ml$^{-1}$) | conc. found (µg ml$^{-1}$) | % found | conc. found (µg ml$^{-1}$) | % found | % found |
| BMB | 0.4 | 0.398 | 99.58 | 0.401 | 100.30 | 99.71 |
| | 0.8 | 0.805 | 100.63 | 0.796 | 99.54 | 100.46 |
| | 1.0 | 0.997 | 99.67 | 1.002 | 100.24 | 99.78 |
| mean | | | 99.96 | | 100.03 | 99.98 |
| ± s.d. | | | 0.58 | | 0.42 | 0.41 |
| $t$-test | | | 0.056 (2.77)[a] | | 0.13 (2.77)[a] | |
| $F$-test | | | 1.97 (19.00)[a] | | 1.04 (19.00)[a] | |
| compound | proposed method | | | | | comparison method [29] |
| | | at 374 nm | | at 384 nm | | |
| | conc. taken (µg ml$^{-1}$) | conc. found (µg ml$^{-1}$) | % found | conc. found (µg ml$^{-1}$) | % found | % found |
| MTK | 0.4 | 0.4017 | 100.43 | 0.4020 | 100.50 | 101.09 |
| | 0.8 | 0.7947 | 99.34 | 0.7941 | 99.26 | 98.30 |
| | 1.0 | 1.0034 | 100.34 | 1.0040 | 100.40 | 100.82 |
| mean | | | 100.04 | | 100.05 | 100.07 |
| ± s.d. | | | 0.61 | | 0.69 | 1.54 |
| $t$-test | | | 0.034 (2.77)[a] | | 0.017 (2.77)[a] | |
| $F$-test | | | 6.47 (19.00)[a] | | 4.99 (19.00)[a] | |

[a]N.B. Each result is the average of three separate determinations.

Limits of detection (LOD) and quantitation (LOQ) were calculated mathematically using ICH equations [39]. LOD were found to be 0.010, 0.017 µg ml$^{-1}$ and 0.052, 0.040 µg ml$^{-1}$, while LOQ were 0.030, 0.050 µg ml$^{-1}$ and 0.158, 0.122 µg ml$^{-1}$ for BMB and MTK, respectively (table 1).

Regarding accuracy, the results obtained from the proposed method were compared with previous published method [29]. No significant difference between the results was noticed as revealed from Student $t$-test and variance ratio $F$-test [40], in table 2.

**Table 5.** Assay results for the determination of the studied drugs in their synthetic mixture by the proposed and comparison method. The figures between parentheses are the tabulated $t$- and $F$-values at $p = 0.05$ [40].

| compound | conc. taken ($\mu g\ ml^{-1}$) | proposed method | | | | | comparison method [29] |
| | | at 284 nm | | at 304 nm | | |
| | | conc. found ($\mu g\ ml^{-1}$) | % found | conc. found ($\mu g\ ml^{-1}$) | % found | % found |
|---|---|---|---|---|---|---|
| BMB | 0.4 | 0.3955 | 98.88 | 0.4024 | 100.60 | 98.84 |
| | 0.8 | 0.8134 | 101.68 | 0.7927 | 99.09 | 101.82 |
| | 1.0 | 0.9911 | 99.11 | 1.0048 | 100.48 | 99.13 |
| mean | | | 99.89 | | 100.06 | 99.93 |
| ± s.d. | | | 1.55 | | 0.84 | 1.64 |
| $t$-test | | | 0.03 (2.77)[a] | | 0.12 (2.77)[a] | |
| $F$-test | | | 1.12 (19.00)[a] | | 3.83 (19.00)[a] | |

| compound | conc. taken ($\mu g\ ml^{-1}$) | proposed method | | | | | comparison method [29] |
| | | at 374 nm | | at 384 nm | | |
| | | conc. found ($\mu g\ ml^{-1}$) | % found | conc. found ($\mu g\ ml^{-1}$) | % found | % found |
|---|---|---|---|---|---|---|
| MTK | 0.4 | 0.4047 | 101.18 | 0.4043 | 101.08 | 99.42 |
| | 0.8 | 0.7858 | 98.23 | 0.7867 | 98.34 | 100.91 |
| | 1.0 | 1.0093 | 100.93 | 1.0086 | 100.86 | 99.56 |
| mean | | | 100.11 | | 100.09 | 99.96 |
| ± s.d. | | | 1.64 | | 1.52 | 0.82 |
| $t$-test | | | 0.14 (2.77)[a] | | 0.13 (2.77)[a] | |
| $F$-test | | | 3.95 (19.00)[a] | | 3.42 (19.00)[a] | |

[a]N.B. Each result is the average of three separate determinations.

The precision of the proposed method was evaluated by testing three concentrations in the same day (intra-day precision) and three successive days (inter-day precision). It was found that the method has good repeatability and intermediate precision as shown in table 3.

Robustness was evaluated by the stability of readings with the minor deliberate changes in experimental parameters. The method did not include buffer or reagent or multi-step sample preparation.

The method selectivity was tested by application of the proposed method to the analysis of tablet formulation containing the two investigated drugs. The additives and excipients did not affect the analysis as indicated by the acceptable percentage recoveries (table 4).

## 3.3. Analysis of synthetic mixtures and combined tablets

Figure 9 represents the second derivative synchronous spectra for three synthetic mixtures of MBM and MTK demonstrating that there is no interference from each other. Moreover, the results obtained were also compared with those of the comparison method [29], and there was no significant difference regarding accuracy and precision [40] as shown in table 5.

## 4. Conclusion

There is a special interest in the analysis of pharmaceutical compounds with decreasing of the analysis time and the solvents need during the analytical process. Therefore, a simple new methodology was

suggested here for the simultaneous determination of bambuterol and montelukast in their combined tablet dosage forms. As the two drugs have native fluorescent properties, the method is based on second derivative synchronous spectrofluorimetric method using water as diluting solvent. The developed method is facile with no need for heating process or organic hazardous solvents. It could be a good alternative in quality control laboratories.

Data accessibility. Data are available at Dryad Digital Repository [41]: Abo El Abass, Samah (2020), Study of different parameters, Dryad, Dataset, https://doi.org/10.5061/dryad.ncjsxksrt.

Authors' contributions. R.E.G. carried out the laboratory work, participated in data analysis, participated in the design of the study and drafted the manuscript. S.A.E.A. carried out the statistical analyses, conceived of the study, designed the study, coordinated the study and submitted the manuscript. H.M.E. participated in data analysis, in the design of the study and drafted the manuscript. All authors gave final approval for publication.

Competing interests. We have no competing interests.

Funding. No funding supported this research.

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
