## [Reviewer comments · Royal Society Open Science]

Review History

RSOS-201156.R0 (Original submission)

Review form: Reviewer 1

Is the manuscript scientifically sound in its present form?

Yes

Are the interpretations and conclusions justified by the results?

Yes

Is the language acceptable?

Yes

Do you have any ethical concerns with this paper?

No

Have you any concerns about statistical analyses in this paper?

No

Recommendation?

Accept with minor revision (please list in comments)

Comments to the Author(s)

- The manuscript represents a spectrofluorimetric method for simultaneous analysis of combined drugs; bambuterol and montelukast. I recommended that the proposed method is rapid and sensitive enough for simultaneous estimation of such mixture. I think it is useful and has advantages over the published methods. The manuscript is accepted in its current form but few points should be explained.

Comments to author:

The proposed method is simple and easily applicable, however some points need to be clarified:

1. The authors mentioned two wavelengths for the determination of each of the studied drugs, Why? And was there a particular difference in sensitivity between the two wavelengths?
2. Page 3, line 82, please change "low analysis time" to "short analysis time"
3. Why did the authors use 2nd derivative and not 1st derivative, please explain!
4. In author contributions: participated should be deleted as it is redundant word

Review form: Reviewer 2 (Azza Rageh)

Is the manuscript scientifically sound in its present form?

Yes

Are the interpretations and conclusions justified by the results?

Yes

Is the language acceptable?

Yes

Do you have any ethical concerns with this paper?

No

Have you any concerns about statistical analyses in this paper?

No

Recommendation?

Accept with minor revision (please list in comments)

Comments to the Author(s)

The manuscript is well-written and topic is interesting. It can be accepted for publication in Royal Society of Open Science after minor revision.

Minor comments

- Rephrase the last point in the highlights part.
- In the introduction section:- - ashma should be corrected.
- The employed concentration of the tested surfactants for example SDS is below its CMC, therefore the authors cannot conclude that surfactants have no impact on the fluorescence intensity. The positive impact of surfactants on fluorescence can be clearly seen when the surfactants are in their micellar forms (above their CMC values).
- Line 82 change overcome into overcomed.

- The last sentence: The method has restricted the use of methanol in the preparation of standard solution and the diluting solvent was water which makes it a green and cost effective method. Should be shorter.
- Line 124: change borate into Borate
- Line 161 : Figures 3& 4 not figure
- Line 204: the brackets of ref 40 are not as in all references
- Reference style isn't unified. Authors names need corrections, please see ref [7], [42] and revise all
- Figure 2 captions: 263 and 286 nm.
- Revise the tabulated values of F-test in Table 2.
- Remove the footnotes for the t- and F-test under Table 3.

Review form: Reviewer 3

Is the manuscript scientifically sound in its present form?

No

Are the interpretations and conclusions justified by the results?

Yes

Is the language acceptable?

Yes

Do you have any ethical concerns with this paper?

No

Have you any concerns about statistical analyses in this paper?

No

Recommendation?

Major revision is needed (please make suggestions in comments)

Comments to the Author(s)

Comments:

1. The title of the manuscript state that the method is simultaneous one. But this is not true. The method can only be used for the determination of the drug in presence of each other in two different measurement using different specifications. However, the word simultaneous means that both drugs can be determined in the same time in single run as in the case of HPLC chromatogram.
2. Page 3, line 83: The authors mentioned that "Furthermore, combining 83 synchronous scanning spectrofluorimetry with derivative techniques provides higher 84 recovery percentages at lower concentration levels," Actually derivative technique could enhance the selectivity of the method but not affect the recovery of the method. In addition higher derivative (specially the third and fourth derivative can severely affect the sensitivity of the method due to the reduction of the amplitude.
3. Page 4, line 25-27: " to reach 134 the highest sensitivity, 25 points were used for deriving the derivative synchronous fluorescence spectra."
4. Page 4, lines 30-32: The authors stated that " The peak amplitudes were measured at 284 & 304 nm for BMB and at 374 & 384 nm for MTK."
5. Page 6, line 50-53: The authors mentioned that "maximum intensity for BMB was obtained at pH 7.5 where for MTK, maximum SFI was at the pH range of 8-9" This means that the fluorescence intensities of both drugs were affected by the pH changes. However the authors after this sentence said, "for both drugs buffer did not enhance the SFI if compared to not using

buffer at all". How could the fluorescence intensities were affected by the pH while the derived synchronous spectra of the same drugs were not affected?

6. Page 20, Figure 2: The figure show the emission spectra of (1) bambuterol after excitation at 263 (2) montelukast after excitation at 286 nm. Also a blank is shown but it is not mentioned at which excitation. The curve should show two additional emission spectra. The first for bambuterol after excitation at 286 (2) montelukast after excitation at 263 nm.
7. Page 21, Fig. 3 and 4: Both figure show that montelukast could be determined in the presence of bambuterol without any interference in the synchronous spectra of bambuterol without the need for derivative estimation.
8. Page 23, figures 7 and 8: The authors did not mention the wavelengths for measuring the fluorescence intensities in the synchronous spectra for both drugs.
9. Page 24, figure 14: The figure could be deleted without affecting the manuscript.
10. According to figure 6 in page 22, the slope for montelukast at 374 nm should be one third of its slope at 384 because the peak amplitude is in this ratio. However, a similar slopes are shown in table 1 (0.219 and 0.241).
11. Table 3 in page 27 should be summarized by deleting the individual values .
12. Table 5 for synthetic tables should deleted as the authors could get the combined commercial tablet dosage forms.

Decision letter (RSOS-201156.R0)

Dear Dr Abo El Abass:

Title: Quick Simultaneous Analysis of Bambuterol and Montelukast Based on Synchronous Spectrofluorimetric Technique.

Manuscript ID: RSOS-201156

The editor assigned to your manuscript has now received comments from reviewers. We would like you to revise your paper in accordance with the referee and Subject Editor suggestions which can be found below (not including confidential reports to the Editor). Please note this decision does not guarantee eventual acceptance.

Please submit your revised paper before 16-Sep-2020. Please note that the revision deadline will expire at 00.00am on this date. If we do not hear from you within this time then it will be assumed that the paper has been withdrawn. In exceptional circumstances, extensions may be possible if agreed with the Editorial Office in advance. We do not allow multiple rounds of revision so we urge you to make every effort to fully address all of the comments at this stage. If deemed necessary by the Editors, your manuscript will be sent back to one or more of the original reviewers for assessment. If the original reviewers are not available we may invite new reviewers.

RSC Associate Editor:
Comments to the Author:
(There are no comments.)

RSC Subject Editor:
Comments to the Author:
(There are no comments.)

Reviewers' Comments to Author:
Reviewer: 1

Comments to the Author(s)
- The manuscript represents a spectrofluorimetric method for simultaneous analysis of combined drugs; bambuterol and montelukast. I recommended that the proposed method is rapid and sensitive enough for simultaneous estimation of such mixture. I think it is useful and has advantages over the published methods. The manuscript is accepted in its current form but few points should be explained.

Comments to author:
The proposed method is simple and easily applicable, however some points need to be clarified:
1. The authors mentioned two wavelengths for the determination of each of the studied drugs, Why? And was there a particular difference in sensitivity between the two wavelengths?
2. Page 3, line 82, please change "low analysis time" to "short analysis time"
3. Why did the authors use 2nd derivative and not 1st derivative, please explain!
4. In author contributions: participated should be deleted as it is redundant word

Reviewer: 2
Comments to the Author(s)
The manuscript is well-written and topic is interesting. It can be accepted for publication in Royal Society of Open Science after minor revision.

Minor comments

- Rephrase the last point in the highlights part.
- In the introduction section:- - ashma should be corrected.
- The employed concentration of the tested surfactants for example SDS is below its CMC, therefore the authors cannot conclude that surfactants have no impact on the fluorescence intensity. The positive impact of surfactants on fluorescence can be clearly seen when the surfactants are in their micellar forms (above their CMC values).
- Line 82 change overcome into overcomed.
- The last sentence: The method has restricted the use of methanol in the preparation of standard solution and the diluting solvent was water which makes it a green and cost effective method. Should be shorter.
- Line 124: change borate into Borate
- Line 161 : Figures 3& 4 not figure
- Line 204: the brackets of ref 40 are not as in all references
- Reference style isn't unified. Authors names need corrections, please see ref [7], [42] and revise all
- Figure 2 captions: 263 and 286 nm.
- Revise the tabulated values of F-test in Table 2.
- Remove the footnotes for the t- and F-test under Table 3.

Reviewer: 3

Comments to the Author(s)

Comments:

1. The title of the manuscript state that the method is simultaneous one. But this is not true. The method can only be used for the determination of the drug in presence of each other in two different measurement using different specifications. However, the word simultaneous means that both drugs can be determined in the same time in single run as in the case of HPLC chromatogram.
2. Page 3, line 83: The authors mentioned that "Furthermore, combining 83 synchronous scanning spectrofluorimetry with derivative techniques provides higher 84 recovery percentages at lower concentration levels," Actually derivative technique could enhance the selectivity of the method but not affect the recovery of the method. In addition higher derivative (specially the third and fourth derivative can severely affect the sensitivity of the method due to the reduction of the amplitude.
3. Page 4, line 25-27: " to reach 134 the highest sensitivity, 25 points were used for deriving the derivative synchronous fluorescence spectra."
4. Page 4, lines 30-32: The authors stated that " The peak amplitudes were measured at 284 & 304 nm for BMB and at 374 & 384 nm for MTK."
5. Page 6, line 50-53: The authors mentioned that "maximum intensity for BMB was obtained at pH 7.5 where for MTK, maximum SFI was at the pH range of 8-9" This means that the fluorescence intensities of both drugs were affected by the pH changes. However the authors after this sentence said, "for both drugs buffer did not enhance the SFI if compared to not using buffer at all". How could the fluorescence intensities were affected by the pH while the derived synchronous spectra of the same drugs were not affected?
6. Page 20, Figure 2: The figure show the emission spectra of (1) bambuterol after excitation at 263 (2) montelukast after excitation at 286 nm. Also a blank is shown but it is not mentioned at which excitation. The curve should show two additional emission spectra. The first for bambuterol after excitation at 286 (2) montelukast after excitation at 263 nm.
7. Page 21, Fig. 3 and 4: Both figure show that montelukast could be determined in the presence of bambuterol without any interference in the synchronous spectra of bambuterol without the need for derivative estimation.
8. Page 23, figures 7 and 8: The authors did not mention the wavelengths for measuring the fluorescence intensities in the synchronous spectra for both drugs.

9. Page 24, figure 14: The figure could be deleted without affecting the manuscript.
10. According to figure 6 in page 22, the slope for montelukast at 374 nm should be one third of its slope at 384 because the peak amplitude is in this ratio. However, a similar slopes are shown in table 1 (0.219 and 0.241).
11. Table 3 in page 27 should be summarized by deleting the individual values .
12. Table 5 for synthetic tables should deleted as the authors could get the combined commercial tablet dosage forms.

Author's Response to Decision Letter for (RSOS-201156.R0)

See Appendices A & B.

RSOS-201156.R1 (Revision)

Review form: Reviewer 1

Is the manuscript scientifically sound in its present form?

Yes

Are the interpretations and conclusions justified by the results?

Yes

Is the language acceptable?

Yes

Do you have any ethical concerns with this paper?

No

Have you any concerns about statistical analyses in this paper?

No

Recommendation?

Accept as is

Comments to the Author(s)

The authors responded to my previous comments

Review form: Reviewer 2 (Azza Rageh)

Is the manuscript scientifically sound in its present form?

Yes

Are the interpretations and conclusions justified by the results?

Yes

Is the language acceptable?

Yes

Do you have any ethical concerns with this paper?

No

Have you any concerns about statistical analyses in this paper?

No

Recommendation?

Accept as is

Comments to the Author(s)

Accept in its current form

Review form: Reviewer 3

Is the manuscript scientifically sound in its present form?

Yes

Are the interpretations and conclusions justified by the results?

Yes

Is the language acceptable?

Yes

Do you have any ethical concerns with this paper?

No

Have you any concerns about statistical analyses in this paper?

No

Recommendation?

Accept with minor revision (please list in comments)

Comments to the Author(s)

1. Page 18, line 24: " to reach the highest sensitivity, 25 points were used for deriving the derivative synchronous fluorescence spectra." Please mention the interval of wavelength in nm rather than number of points or make the suitable change to clarify the meaning of the number of points.
2. By using the values of the slope and intercept in table 1, the calculated value of peak amplitude for montelukast at 374 and 384 were found to be 0.425 and 0.556, respectively. The calculated value at 374 is close to that in Figure 6, However the calculated value at 384 is very low (-0.556) compared to the value of the blue curve in Figure 6 (about -0.915). Therefore, the values of the intercept and slope should be corrected.
3. The title of the vertical axis in Fig. 5, 6 and 9 should be changed to peak amplitude Similarly The title of the vertical axis in Fig. 7 and 8 should be changed to FI (fluorescence intensity).
4. Why the tabulated values of F-test is not the same although the same number of determinations are used in all cases?

Decision letter (RSOS-201156.R1)

Dear Dr Abo El Abass:

Title: Quick Simultaneous Analysis of Bambuterol and Montelukast Based on Synchronous Spectrofluorimetric Technique.
Manuscript ID: RSOS-201156.R1

Thank you for submitting the above manuscript to Royal Society Open Science. On behalf of the Editors and the Royal Society of Chemistry, I am pleased to inform you that your manuscript will be accepted for publication in Royal Society Open Science subject to minor revision in accordance with the referee suggestions. Please find the reviewers' comments at the end of this email.

The reviewers and handling editors have recommended publication, but also suggest some minor revisions to your manuscript. Therefore, I invite you to respond to the comments and revise your manuscript.

Because the schedule for publication is very tight, it is a condition of publication that you submit the revised version of your manuscript before 08-Oct-2020. Please note that the revision deadline will expire at 00.00am on this date. If you do not think you will be able to meet this date please let me know immediately.

Kind regards,
Dr Laura Smith
Publishing Editor, Journals

RSC Associate Editor:
Comments to the Author:
(There are no comments.)

RSC Subject Editor:
Comments to the Author:
(There are no comments.)

Reviewer comments to Author:
Reviewer: 2

Comments to the Author(s)
Accept in its current form

Reviewer: 1

Comments to the Author(s)
The authors responded to my previous comments

Reviewer: 3

Comments to the Author(s)
1. Page 18, line 24: " to reach the highest sensitivity, 25 points were used for deriving the derivative synchronous fluorescence spectra." Please mention the interval of wavelength in nm rather than number of points or make the suitable change to clarify the meaning of the number of points.

2. By using the values of the slope and intercept in table 1, the calculated value of peak amplitude for montelukast at 374 and 384 were found to be 0.425 and 0.556, respectively. The calculated value at 374 is close to that in Figure 6, However the calculated value at 384 is very low (-0.556) compared to the value of the blue curve in Figure 6 (about -0.915). Therefore, the values of the intercept and slope should be corrected.
3. The title of the vertical axis in Fig. 5, 6 and 9 should be changed to peak amplitude Similarly The title of the vertical axis in Fig. 7 and 8 should be changed to FI (fluorescence intensity).
4. Why the tabulated values of F-test is not the same although the same number of determinations are used in all cases?

Author's Response to Decision Letter for (RSOS-201156.R1)

See Appendix C.

Decision letter (RSOS-201156.R2)

Dear Dr Abo El Abass:

Title: Quick Simultaneous Analysis of Bambuterol and Montelukast Based on Synchronous Spectrofluorimetric Technique.
Manuscript ID: RSOS-201156.R2

It is a pleasure to accept your manuscript in its current form for publication in Royal Society Open Science. The chemistry content of Royal Society Open Science is published in collaboration with the Royal Society of Chemistry.

RSC Associate Editor
Comments to the Author:
(There are no comments.)

Reviewer(s)' Comments to Author:

Appendix A

Dr. Samah Abo El Abass

Mansoura University,
Faculty of Pharmacy,
Pharm. Analytical Chemistry Dept
Mansoura 35516
Egypt.
Office: +20502246253
Fax : +20502247496

Mansoura, 23.04.2019

Dear Prof. Editor of Royal Society Open Science,

Herewith, I submit the manuscript **“Quick Simultaneous Analysis of Bambuterol and Montelukast Based on Synchronous Spectrofluorimetric Technique.”**(Authors: Rania El Gamal, Samah Abo El Abass and Heba M. Elmansi) as a full-length article for publication in your esteemed journal.

This is the first report on the simultaneous spectrofluorimetric determination of bambuterol (BMB) and montelukast (MTK). The studied drugs are present in combined tablet formulations with a pharmaceutical ratio of 1:1 to treat chronic asthma, bronchospasm, and reversible airway obstruction. Scanning the fluorescence spectra of the studied drugs at their optimum excitation/emission revealed that their spectra suffered from significant overlapping that hinders their simultaneous determination. Accordingly, a simple new method was investigated in order to analyze both BMB and MTK simultaneously without interference from each other relying on derivative synchronous spectrofluorimetry. The developed method is characterized by being highly sensitive, very fast, simple and precise. The work is new and original and not under consideration elsewhere. There is no conflict of interest.

Best Regards

Yours sincerely,

Dr. Samah Abo El Abass

Mansoura University,
Faculty of Pharmacy,
Pharm. Analytical Chemistry Department.

Appendix B

Reply to reviewer's comments

Manuscript ID: RSOS-201156

Title: Quick Simultaneous Analysis of Bambuterol and Montelukast Based on Synchronous Spectrofluorimetric Technique.

-The manuscript was carefully revised and all the comments of the reviewers were carefully considered in preparing the revised version.

For Reviewer 1:

1. The authors mentioned two wavelengths for the determination of each of the studied drugs, Why? And was there a particular difference in sensitivity between the two wavelengths?

- BMB can be determined at 284 & 304 nm in presence of MTK, and MTK could be determined at 374& 384 nm in the presence of BMB. The proposed method can be applied at two wavelengths as shown in figures 5&6. For each drug the sensitivity at both wavelengths was almost the same, however, they were both included in the manuscript to help in quality control analysis of the mixture where at certain laboratory the instrument specification may vary, hence any of the specified wavelengths can be utilized with high sensitivity and accuracy, which in turn could be considered as an advantage of the method.

2- Page 3, line 82, please change "low analysis time" to "short analysis time"

- It has been changed as recommended.

3- Why did the authors use 2nd derivative and not 1st derivative, please explain!

- The proposed method was based on applying second derivative amplitude for both BMB & MTK. The first derivative was tried; it was successful for determination of BMB in presence of MTK but not the opposite, hence, 2nd derivative was tried, where it was possible to analyze each of the studied drugs in presence of the other without any interference.

4- In author contributions: participated should be deleted as it is redundant word.

- The repeated word "participated" has been removed.

For Reviewer 2:

- Rephrase the last point in the highlights part.

- It has been corrected.

- In the introduction section: ashma should be corrected.

- It has been corrected.

- The employed concentration of the tested surfactants for example SDS is below its CMC, therefore the authors cannot conclude that surfactants have no impact on the fluorescence intensity. The positive impact of surfactants on fluorescence can be clearly seen when the surfactants are in their micellar forms (above their CMC values).

- The effect of different surfactants as organized medium has been studied using different organized media. We use the concentration of 0.1 % of each surfactant because it gave low blank reading. Also, we tried to higher concentration up to (0.5 % of SDS) to reach to critical micelle concentration but the blank background was high and diminishes the fluorescence enhancement.

We follow the procedures as cited in most of the published articles for fluorescence enhancement by surfactants use low concentrations (up to 1%) to avoid the high blank reading like:

*Walash, M.I., Metwally, M.E., Eid, M., El-Shaheny, R.N. Validated spectrophotometric methods for determination of Alendronate sodium in tablets through nucleophilic aromatic substitution reactions. Chem. Cent. J. 2012

*Ibrahim, F.A., El-Enany, N., El-Shaheny, R.N., Mikhail, I.E. Simultaneous determination of desloratadine and montelukast sodium using second-derivative synchronous fluorescence spectrometry enhanced by an organized medium with applications to tablets and human plasma. *Luminescence*. 2015 Jun;30(4):485-94

*F. Belal, M. K. Sharaf El-Din, M. M. Tolba, H. Elmansi. Micelle-Enhanced Spectrofluorimetric Method for Determination of Cyproheptadine Hydrochloride in Tablets: Application to In-Vitro Drug Release and Content Uniformity Test. Journal of Fluorescence, 2014, 24: 85-91.

- The mechanism of the enhancement of the fluorescence of the studied drugs in presence of different surfactants is probably due to a very rigid microenvironment, which is capable of restricting the freedom of fluorophores and consequently diminishes the probabilities of non-radiative processes and provides a relatively high viscous microenvironment that can inhibit quenching by molecular oxygen. These factors might increase the fluorescence quantum yield and enhance the fluorescence signals **according to the reference:**

Andrade-Eiroa A, De-Armas G, Estela JM, Cerda V. Critical approach to synchronous spectrofluorimetry. Trends Anal Chem, 2010; 29: 885–901

- Line 82 change overcome into overcome.

- It has been corrected.

- The last sentence: The method has restricted the use of methanol in the preparation of standard solution and the diluting solvent was water which makes it a green and cost effective method. Should be shorter.

- It has been changed as recommended.

-Line 124: change borate into Borate.

- It has been changed as recommended.

- Line 161: Figures 3& 4 not figure.

- It has been changed as recommended.

- Line 204: the brackets of ref 40 are not as in all references.

- It has been changed as recommended.

- Reference style isn't unified. Authors names need corrections, please see ref [7], [42] and revise all.

- It has been revised and corrected as recommended.

- Figure 2 captions: 263 and 286 nm.

- These are the excitation wavelengths for optimal emission of each of the studied drugs.

- Revise the tabulated values of F-test in Table 2.

- It has been revised as recommended.

- Remove the footnotes for the t- and F-test under Table 3.

- It has been removed as recommended.

For Reviewer 3:

1. The title of the manuscript state that the method is simultaneous one. But this is not true. The method can only be used for the determination of the drug in presence of each other in two different measurement using different specifications. However, the word simultaneous means that both drugs can be determined in the same time in single run as in the case of HPLC chromatogram.

_ I respect your opinion, however they are not two different measurements, it is a single one but quantitative measurement is done at different wavelengths, the situation is that a mixture of the studied drugs is run in synchronous measurement mode then the 2nd derivative curve is derived from the zero synchronus one.

2. Page 3, line 83: The authors mentioned that "Furthermore, combining 83 synchronous scanning spectrofluorimetry with derivative techniques provides higher 84 recovery percentages at lower concentration levels," Actually derivative technique could enhance the selectivity of the method but not affect the recovery of the method. In addition, higher derivative (specially the third and fourth derivative can severely affect the sensitivity of the method due to the reduction of the amplitude.

- The selectivity of an analytical method is its ability to determine a particular compound in presence of other components, if there is an encountered interference from other components with an increase or decrease of the parameter measured in the utilized technique, this would in turn results in a concomitant increase or decrease in recovery when compared to a pure form of the analyte.

This is why sometimes interference from additives in dosage form analysis for example could result in higher % recovery if compared to authentic drug.

For higher derivative (specially the third and fourth derivative) yes, they can severely affect the sensitivity of the method due to the reduction of the amplitude, however we usually compromise factors to attain optimal conditions, and mostly these higher derivatives are not utilized except when it is not possible to determine the analyzed samples by lower ones.

3. Page 4, line 25-27: " to reach 134 the highest sensitivity, 25 points were used for deriving the derivative synchronous fluorescence spectra."

- The No. of points greatly influences the shape of the derivative synchronous fluorescence spectra, that has a great impact on accurate measurement of the SFI amplitude and hence the sensitivity.

4. Page 4, lines 30-32: The authors stated that " The peak amplitudes were measured at 284 & 304 nm for BMB and at 374 & 384 nm for MTK."

- Yes, you are right each of the two drugs was determined at 2 wavelengths, with almost the same sensitivity.

5. Page 6, line 50-53: The authors mentioned that "maximum intensity for BMB was obtained at pH 7.5 where for MTK, maximum SFI was at the pH range of 8-9" This means that the fluorescence intensities of both drugs were affected by the pH changes. However the authors after this sentence said, "for both drugs buffer did not enhance the SFI if compared to not using buffer at all". How could the fluorescence intensities were affected by the pH while the derived synchronous spectra of the same drugs were not affected?

- The effect of buffer was studied utilizing different buffer with different pH values ranging from (3.6-9), as mentioned in the manuscript pH 7.5 for BMB and pH 8-9 for MTK showed the maximal fluorescence intensities for the studied drugs in buffer, but when comparing the values of these maximal fluorescence intensities of drugs to the SFI measured in absence of buffer, the net conclusion is that even the best buffer conditions were inferior to not using buffer at all. This is the reason why the method was carried out without using any buffer.

Hopefully the point is clarified....

6- Page 20, Figure 2: The figure shows the emission spectra of (1) bambuterol after excitation at 263 (2) montelukast after excitation at 286 nm. Also a blank is shown but it is not mentioned at which excitation. The curve should show two additional emission spectra. The first for bambuterol after excitation at 286 (2) montelukast after excitation at 263 nm.

- The selected wavelengths were the maximum excitation wavelengths for both drugs, a trial to scan each of them at the maximum excitation of the other also resulted in an overlap as the demonstrated one in the figure but with a slight shift of peaks, hence it was not included to avoid confusion.
- As for the blank since the blank is water the emission spectrum of it was almost the same when switching excitation from 263 to 286.

7. Page 21, Fig. 3 and 4: Both figure show that montelukast could be determined in the presence of bambuterol without any interference in the synchronous spectra of bambuterol without the need for derivative estimation.

- This is totally right, but this could determine only MTK but not BMB this is why 2nd derivative synchronous was utilized to determine both drugs simultaneously each one in the presence of the other without any interference. This is mostly clear in figure 9 that shows the determination of mixture of both drugs by the specified method.

8. Page 23, figures 7 and 8: The authors did not mention the wavelengths for measuring the fluorescence intensities in the synchronous spectra for both drugs.

- The wavelengths for measuring the SFI were 388 nm for montelukast and 287 nm for bambuterol, these were added to figures 7 & 8 and to figure caption.

9. Page 24, figure 14: The figure could be deleted without affecting the manuscript.

The No. of figures are 9. There is no figure 14.

But if you mean figure 9, This figure shows the effectiveness of the method in determining both drugs in mixture and hence in their commercially available dosage form at their specified ratio.

10. According to figure 6 in page 22, the slope for montelukast at 374 nm should be one third of its slope at 384 because the peak amplitude is in this ratio. However, a similar slope is shown in table 1 (0.219 and 0.241).

- This concept is right if the intercept is the same, however as shown in table 1, the intercept is different.

11. Table 3 in page 27 should be summarized by deleting the individual values.

- Done as per your recommendation

12. Table 5 for synthetic tables should be deleted as the authors could get the combined commercial tablet dosage forms.

- Table 5 represents the data for a laboratory prepared mixture of both drugs in their pure forms without any additives not a synthetic tablet, the data obtained were used to determine the recovery for tablet formulations.

Appendix C

Reply to reviewer's comments

Manuscript ID: RSOS-201156.R1

Title: Quick Simultaneous Analysis of Bambuterol and Montelukast Based on Synchronous Spectrofluorimetric Technique.

-The manuscript was carefully revised and all the comments of reviewer 3 were carefully considered in preparing the revised version.

For Reviewer 3:

1. Page 18, line 24: " to reach the highest sensitivity, 25 points were used for deriving the derivative synchronous fluorescence spectra." Please mention the interval of wavelength in nm rather than number of points or make the suitable change to clarify the meaning of the number of points.

- The interval of wavelength, $\Delta\lambda$ is already studied and discussed (page 6, lines 172-175), however the no of points is another studied factor that affects the synchronous spectra shapes, where the optimal number of points improves the spectra shapes which impact proper intensity recording that in turn impacts the sensitivity of method.

2. By using the values of the slope and intercept in table 1, the calculated value of peak amplitude for montelukast at 374 and 384 were found to be 0.425 and 0.556, respectively. The calculated value at 374 is close to that in Figure 6, However the calculated value at 384 is very low (-0.556) compared to the value of the blue curve in Figure 6 (about -0.915). Therefore, the values of the intercept and slope should be corrected.

- I respect your sharp notice, but actually this is true if we are measuring at peaks maxima, however if you magnify the figure, and checked the measurement is done at the zero crossing point with the BMT spectrum and this comes a bit to the left from the maxima where the measurements were done.

3. The title of the vertical axis in Fig. 5, 6 and 9 should be changed to peak amplitude
Similarly The title of the vertical axis in Fig. 7 and 8 should be changed to FI
(fluorescence intensity).

- The title of the axis has been changed as recommended.

4. Why the tabulated values of F-test is not the same although the same number
of determinations are used in all cases?

- The tabulated values of F-test have been revised well. The tabulated value of F-test is based on standard deviation value for the proposed and the comparison method not only on the number of measurements. It is calculated by using the higher standard deviation is the numerator and the lower standard deviation is the denominator even that the number of measurements are the same. So in one value the tabulated F value is for 4/2 and in other case is 2/4 according to the higher in standard deviation.